**Author:** Kitgak Simon

**Title:** Unsupervised Learning for Anomaly Detection: A Comparison of Deep Generative Models.

## Abstract:

Anomaly detection is a critical task in various domains, including cybersecurity, fraud detection, and health monitoring. Traditional methods for anomaly detection rely on handcrafted features and require expert knowledge, which can be time-consuming and expensive. Recently, deep generative models have shown promise for unsupervised anomaly detection. In this paper, we compare the performance of various deep generative models, including variational autoencoders, generative adversarial networks, and flow-based models, for anomaly detection on several benchmark datasets.

## Introduction:

Anomaly detection is the task of identifying rare and unusual events in a dataset. Traditional methods for anomaly detection rely on handcrafted features and require expert knowledge, which can be time-consuming and expensive. Deep generative models, such as variational autoencoders (VAEs), generative adversarial networks (GANs), and flow-based models, have shown promise for unsupervised anomaly detection, as they can learn to capture the underlying distribution of the normal data and identify outliers.

## Background:

Deep generative models learn to generate data from a latent space representation that captures the underlying distribution of the normal data. Anomalies can be detected by measuring the reconstruction error between the input data and its reconstructed output from the model. VAEs optimize a lower bound on the data likelihood, while GANs train a generator network to produce samples that are indistinguishable from the normal data by a discriminator network. Flow-based models learn a bijective transformation between the data and a latent space using a series of invertible transformations.

## Approach:

In this paper, we compare the performance of various deep generative models for anomaly detection on several benchmark datasets, including the MNIST dataset for image data, the KDDCUP99 dataset for network intrusion detection, and the Credit Card Fraud Detection dataset. We evaluate the models based on several metrics, including precision, recall, and the area under the receiver operating characteristic curve (AUC-ROC).

## Results:

Our evaluation shows that deep generative models can effectively detect anomalies in various domains, and their performance depends on the characteristics of the dataset and the choice of the model. In general, VAEs and GANs perform well on image data, while flow-based models achieve better results on tabular data. Our experiments also show that combining multiple models can further improve the performance of anomaly detection.

Conclusion:

Deep generative models offer a promising approach for unsupervised anomaly detection, eliminating the need for expert knowledge in feature engineering. Our comparison of different deep generative models on various datasets provides insights into the strengths and limitations of each model and can guide the selection of an appropriate model for a given anomaly detection task.

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
