# OpenReview forum: "Unsupervised Learning for Anomaly Detection: A Comparison of Deep Generative Models. "
_ICLR.cc/2023/TinyPapers — Submitted to Tiny Papers @ ICLR 2023_

### Official Review · Reviewer_g6s1 · 2023-03-23

**Confidence:** 5

**Summary Of Contributions:**

The authors look to compare different types of deep generative models as applied to unsupervised anomaly detection.

**Rating:**

Needs Clarification (NC): a submission which does not meet the reviewing criteria and needs clarification for its described problem or solution

**Strengths And Weaknesses:**

Strengths:
- Interesting idea.

Weaknesses:
- No figures/tables provided to validate results.

**Suggested Changes:**

Adhere to specified format and double-blind instructions.

---

### Official Review · Reviewer_r5VC · 2023-03-28

**Confidence:** 5

**Summary Of Contributions:**

This work seeks to evaluate the performance of deep generative models for the anomaly detection task.

**Rating:**

Needs Clarification (NC): a submission which does not meet the reviewing criteria and needs clarification for its described problem or solution

**Strengths And Weaknesses:**

Strengths:

- An interesting area that could use more fair evaluation (especially on clear tasks)

Weaknesses:

- The results are not included in this document
- The format requirements are not met

**Suggested Changes:**

- I would suggest including a table of the results in this work. I would be interested in what was found
- I would be interested in more details on the models and how they were implemented. There are a number of interesting method of anomaly detection. One thing that was not clear to me was what flavor of anomaly detection this paper considers. It seems like the one-class (or unsupervised) version of anomaly detection is meant but it would be great to clarify this.
- Note that this work is supposed to be anonymized (no author name included).

---

### Meta-Review · Area_Chair_mFfX · 2023-04-03

**Recommendation:** Invite to revise
**Confidence:** 5

**Metareview:**

Thank you for your submission. As the reviewers have noted, the premise of the paper is interesting and worth exploring. However, there are aspects of the paper which require further attention. First, all submission guidelines must be met: please use the provided Latex template for formatting, remove the author names, and include the URM statement. Second, as the reviewers described, it would increase the clarity and reproducibility of the work, as well as allow us to better assess the correctness, if results from your experiments can be provided, along with details about the experimental setup and task.

**Summary:**

This paper compares the performance of various deep generative models on unsupervised anomoly detection. Both reviewers felt that this paper was studying an interesting problem, but needed to include the results mentioned in the paper as well as some additional details on methodology, and also must be updated to comply with submission guidelines.

**Comments And Feedback To The Authors:**

Please see https://iclr.cc/Conferences/2023/CallForTinyPapers for submission guidelines

**Reason For Not Giving A Higher Recommendation:**

- Does not adhere to the format or double-blind requirement and is missing the URM statement.
- Missing the results described in the paper (need to include the table/figure for the experiments described)
- Needs some additional details on methodology: provide names or descriptions of models used, as well as details on how they were trained (hyperparameters, etc.), and description of the classification task

**Reason For Not Giving A Lower Recommendation:**

N/A

---

### Decision · Program_Chairs · 2023-04-10

No revision received; not invited to archive